# Paper microfluidic implementation of loop mediated isothermal amplification for early diagnosis of hepatitis C virus

Weronika Witkowska McConnell [1,2], Chris Davis[1], Suleman R. Sabir[1], Alice Garrett[2], Amanda Bradley-Stewart[3], Pawel Jajesniak[2], Julien Reboud [2], Gaolian Xu[2], Zhugen Yang [2], Rory Gunson[3], Emma C. Thomson [1✉] & Jonathan M. Cooper [2✉]

The early diagnosis of active hepatitis C virus (HCV) infection remains a significant barrier to the treatment of the disease and to preventing the associated significant morbidity and mortality seen, worldwide. Current testing is delayed due to the high cost, long turnaround times and high expertise needed in centralised diagnostic laboratories. Here we demonstrate a user-friendly, low-cost pan-genotypic assay, based upon reverse transcriptase loop mediated isothermal amplification (RT-LAMP). We developed a prototype device for point-of-care use, comprising a LAMP amplification chamber and lateral flow nucleic acid detection strips, giving a visually-read, user-friendly result in <40 min. The developed assay fulfils the current guidelines recommended by World Health Organisation and is manufactured at minimal cost using simple, portable equipment. Further development of the diagnostic test will facilitate linkage between disease diagnosis and treatment, greatly improving patient care pathways and reducing loss to follow-up, so assisting in the global elimination strategy.

---

[1] MRC-University of Glasgow, Centre for Virus Research, University of Glasgow, Glasgow, UK. [2] Division of Biomedical Engineering, James Watt School of Engineering, University of Glasgow, Glasgow, UK. [3] West of Scotland Specialist Virology Centre, New Lister Building, Glasgow Royal Infirmary, Glasgow, UK. ✉email: emma.thomson@glasgow.ac.uk; jon.cooper@glasgow.ac.uk

Hepatitis C virus (HCV) is a major cause of liver-related mortality with 71 million people chronically infected globally. Recent advances in direct acting antiviral treatments have improved cure rates to >95%. However, currently an estimated 80% of all infected individuals are unaware of their status due to the asymptomatic nature of infection. Many of these patients will remain undiagnosed until irreversible clinical manifestations, such as liver cirrhosis and hepatocellular carcinoma, develop, contributing to the 400,000 HCV related deaths reported every year[1].

The World Health Organisation (WHO) has established a global elimination strategy to reach significant disease reduction targets by 2030[2,3]. The most challenging obstacle in achieving this ambitious goal is the rapid diagnosis of patients and their integration into an appropriate clinical care pathway for treatment[3].

Current diagnostic strategies rely on testing for anti-HCV antibodies followed by RNA or core antigen detection[4]. This two-step process requires a centralised laboratory infrastructure and delays active HCV infection diagnosis by significant periods of time. Successful implementation is also limited by cost, long turnaround times and the high level of expertise required for diagnostic testing. High risk groups, including people-who-inject-drugs (PWID), are often lost to follow-up due to the multiple visits required for HCV diagnosis. Additionally, absent or delayed seroconversion, particularly in immunocompromised patients, may reduce overall rates of diagnosis[5].

Most individuals with HCV reside in low and middle-income countries (LMICs) with only limited access to diagnosis[1,6]. Although recent advances in HCV nucleic acid amplification tests (NAATs) have provided a possible solution for improving the management of HCV diagnosis, the existing diagnostic platforms still pose several limitations, including the high cost and training requirements associated with polymerase chain reaction (PCR) assays, especially when used in decentralised testing[7,8]. Not only are simpler and cheaper NAATs for point-of-care (POC) testing required in order to improve HCV diagnosis and eliminate the need for follow-up visits[3], but importantly, the selection of a universal NAATs' target is needed. HCV, like many RNA viruses, exhibits high genetic diversity, with eight distinct HCV genotypes and at least 90 different subtypes identified with varied prevalence globally[9,10].

Loop mediated isothermal amplification (LAMP) assays provide high sensitivity and specificity through the use of 4–6 primers, which target 6–8 regions within a sequence of interest[11,12]. The amplification reaction takes place at a constant temperature between 60–65 °C, offering a cheaper alternative to the traditional PCR assays, with minimal equipment requirements, which has resulted in its use in decentralised (POC) settings to detect the presence of a variety of pathogens[13–15].

Recently, LAMP assays have also been developed for HCV detection in centralised facilities. Colorimetric outputs have also been developed to enable the detection without bulky optical instrumentation with excellent performance for some genotypes (e.g. Hongjaisee S. et al.[16] showed 100% sensitivity with genotype 6). However, these assays showed limited diagnostic performance when used with varied genotypes or low viral loads[17–19]. Their application as POC tests is thus restricted, especially in LMICs[20], where a wide variation in the genetic diversity may lead to a lower efficacy of tests.

In this study, we now demonstrate a pan-genotypic HCV LAMP assay, which we validated in a prototype POC diagnostic device as a paper-microfluidic, visually read lateral flow test. The assay was validated as part of a double-blind clinical study of samples from patients with a range of viral loads and genotypes as determined by a highly sensitive gold-standard Abbott RealTime RT-PCR HCV test, within a clinical reference laboratory and our own in-house HCV RT-PCR. We subsequently implemented the LAMP nucleic acid detection strip method into a portable and user-friendly device, with an easy-to-use readout for target detection.

## Results

**Optimisation of HCV LAMP primers**. Previously published LAMP primers were selected based on evidence of a low limit of detection from HCV RNA (50 IU/mL)[17]. In this study, the use of an additional accelerating primer (AP, Table 1) ensured improved sensitivity and specificity across several HCV genotypes. We analysed an alignment file of over 200 sequences of major HCV genotypes and subtypes[21]. The majority of the observed mismatches were in the middle of the primers (Fig. S1). We only noted one mismatch within the last base pair of the 3′ end of the backward loop primer (BLP), within genotype 3 sequences. Focussing on genotype 3 and in order to further improve the previously published assay, we removed the cytosine mismatch at the 3′ end of BLP. Additionally, a cytosine was added at the beginning of the primer, in order to conserve the primer melting temperature (Fig. S1 and Table 1). The removal of the cytosine mismatch and the subsequent use of the new primer improved the time to positivity by 21% in genotype 3 (Fig. S2). As a confirmation, the new BLP was also compared to the original BLP performance on genotype 1 targets, which, did not result in significant differences in the assay performance as there were no mismatches within this genotype.

**Clinical sensitivity and specificity**. Using optimised primers, we conducted a double-blind study of LAMP on RNA and cDNA from samples with a range of viral loads and from different HCV genotypes, including a recently identified genotype 7a (Table 2)[22].

The performance of HCV LAMP using RNA samples (RT-LAMP) directly and HCV LAMP using cDNA as template were compared with a highly sensitive in-house HCV RT-PCR by genotype and viral load (Tables 2 and 3). The in-house RT-PCR assay has been characterised previously as a quantitative assay (qRT-PCR[23]).

HCV RT-LAMP and HCV LAMP assays detected 96 and 97 HCV samples out of 100 respectively, comparable with the RT-PCR assay which detected 96/100 samples. The false negatives correlated well between the three assays (in two cases false negatives were due to low viral loads, ≤3.95 $\log_{10}$ IU/mL). RT-LAMP failed to detect a single genotype 4 sample with high viral load (5.64 $\log_{10}$ IU/mL). RT-LAMP and LAMP assays both exhibited high specificity, correctly identifying 91 and 90 control

---

**Table 1 LAMP primer sequences.**

| Primer name | Sequence (5′ → 3′) |
|---|---|
| F3 | ACT CCA CCA TGA ATC ACT C |
| B3 | ATC AGG CAG TAC CAC AAG G |
| FIP | AGG CTG YAC GAC ACT CAT AC-CTG TGA GGA ACT ACT GTC TTC |
| BIP | GGA TMA ACC CRC TCA ATG CC-TCG CRA CCC AAC RCT AC |
| FLP | GCC ATG GCT AGA CGC T |
| BLP | **C**GT GCC CCC GCR AGA[**C**] |
| AP | <u>T</u>TC CGC AGA CCA CTA TGG CTC T |
| FLP BIO | [BIO] GCC ATG GCT AGA CGC T |
| BLP FITC | [FITC] CGT GCC CCC GCR AGA |

Primer sequences were as published by Yang et al.[17], with modifications marked in bold, underlined. The bracket shows the original cytosine position.
*FIP* forward internal primer, *BIP* backward internal primer, *FLP* forward loop primer, *BLP* backward loop primer, *AP* accelerating primer, *Bio* Biotin, *FITC* fluorescein isothiocyanate.

**Table 2 Sample detection by genotype and viral load.**

| Genotype | RT-LAMP | LAMP | RT-PCR | Total samples tested (%) |
|---|---|---|---|---|
| 1 | 25 | 25 | 25 | 26 (26%) |
| 2 | 14 | 14 | 14 | 14 (14%) |
| 3 | 22 | 22 | 21 | 23 (23%) |
| 4 | 21 | 22 | 22 | 22 (22%) |
| 5 | 3 | 3 | 3 | 3 (3%) |
| 6 | 1 | 1 | 1 | 1 (1%) |
| 7 | 1 | 1 | 1 | 1 (1%) |
| Unknown | 9 | 9 | 9 | 10 (10%) |
| Viral load (log$_{10}$ IU/mL) | | | Total | 100 |
| 1.7–3.95 | 3 | 3 | 2 | 5 (5%) |
| 4.4–4.85 | 12 | 12 | 12 | 12 (12%) |
| 5.2–5.98 | 36 | 37 | 37 | 37 (37%) |
| 6.05–6.97 | 37 | 37 | 37 | 37 (37%) |
| Unknown | 8 | 8 | 8 | 9 (9%) |
| | | | Total | 100 |

Experiments were performed in duplicate and at least one positive replicate was interpreted as a positive result.

**Table 3 Sensitivity and specificity of HCV assays.**

| Method | RT-LAMP | LAMP | RT-PCR |
|---|---|---|---|
| True positive | 96 | 97 | 96 |
| False negative | 4 | 3 | 4 |
| Total (sensitivity) | 100 (96%) | 100 (97%) | 100 (96%) |
| True negative | 91 | 90 | 100 |
| False positive | 9 | 10 | 0 |
| Total (specificity) | 100 (91%) | 100 (90%) | 100 (100%) |

Abbott RealTime HCV assay (RT-PCR) was used as the gold standard.

samples out of 100 respectively, although our in-house RT-PCR assay had a higher specificity with no false positives.

**Time to detection**. In order to evaluate the impact of genotype and viral load on the performance of both HCV RT-LAMP and LAMP assay, we defined samples as positive when the fluorescence signal was at least ten standard deviations above the mean baseline fluorescence of the positive control. The time to positive was then determined as described in Fig. S3. The majority of positive samples were detected within 30 min (Fig. 1a, b). For RT-LAMP, genotype 3 and 4 detection took longer than genotype 1 and 2. Similarly, for cDNA, genotype 3 positivity occurred later than all other genotypes. There was an inverse relationship between time to detection and viral load and significant differences occurred between the 1.70–3.95 log$_{10}$ IU/mL group, 4.40–4.85 log$_{10}$ IU/mL group and the 6.05–6.97 log$_{10}$ IU/mL group for both RNA and cDNA (Fig. 1c, d).

Since most samples were detected within the first 30 min, we assessed if this would be a suitable detection time period. Receiver operating characteristic (ROC) curves for both LAMP and the RT-PCR were used to determine the efficacy by which the assays would distinguish between HCV-positive and control samples at different time points or cycles (Fig. 2, Fig. S4 and Tables S1–2). Both RT-LAMP and LAMP had areas under the curve of 0.97 (Fig. 2A, B), showing close statistical similarity to the RT-PCR assay (Fig. S4 and Table S1). The cut-off of <29 min for RT-LAMP was equivalent to 95.0% sensitivity (95% CI: 88.7–98.4%) and 94.0% specificity (95% CI: 87.4–97.8%) (Fig. 2C and Table S2). LAMP had the same sensitivity at <29.75 min, but the

specificity was slightly lower—92.0% (95% CI: 84.8–96.5%) (Fig. 2D and Table S3). To achieve the specificity of 98%, recommended by the WHO, (98.0%, 95% CI of 93.0–99.8%), the cut-off value would be set to <24.75 min for RT-LAMP and <26.75 min for LAMP, resulting in a sensitivity of 90.0% (95% CI of 82.4–95.1%) and 92.0% (95% CI of 84.8–96.5%) for RT-LAMP and LAMP, respectively (Fig. 2C, D, Tables S1–3)[24].

**Analytical sensitivity and end-point detection**. Analytical sensitivity of HCV LAMP was assessed using serial dilutions of plasmids containing HCV sub-genomic replicon over 40 min by three different detection methods; nucleic acid detection strips, gel electrophoresis and changes in fluorescence over time. The concentration of HCV ranged between 2.6 log$_{10}$ copies/reaction and 4.1 log$_{10}$ copies/reaction. Figure 3 shows the mechanism of strip detection (Fig. 3a), the assembly of the lateral flow device (Fig. 3b), and the results from the analytical sensitivity experiments (Fig. 3c). The lower limit of detection was the same for each method, below 2.6 log$_{10}$ copies/reaction (broadly equivalent to a detection threshold of 398 copies/reaction, which is lower than the recommended ≤3000 IU/mL limit of detection by WHO)[2].

The nucleic acid detection strips used in this study offered an easy-to-interpret pregnancy test-like result with two bands for a positive reaction and one (control) band for negative. The gel electrophoresis revealed a characteristic smear with ladder-like banding pattern for all samples except the negative control. The size of the initial bands (274 bp) correlate with the region between F3/B3 representing the initial stem loop formation of the reaction. As the reaction proceeded, larger constructs were created, with random termination, generating a other bands with a smear-like pattern[11,12,17]. The results correlated with the nucleic acid detection strips. There was an inverse relationship between viral load and time to positivity and all positive reactions were statistically different from the negative control ($p < 0.0001$). The double-blind study revealed that optimal sensitivity and specificity occur at <30 min with a lower limit of detection of 417 IU/mL.

Nucleic acid detection strips are easy to visualise and interpret and are known to correlate well with other detection methods and were thus used as the basis for the development of a low-cost, microfluidic HCV test, which could be used either in bedside and/ or POC settings. An enclosed lateral flow device containing a LAMP reaction chamber, valves and detection strips was manufactured in-house using methods adapted from previous studies (shown in Fig. 3b and Fig. S5)[15]. The LAMP chamber was incubated in a heat block for 30 min at 65 °C by inserting the device upright (thus ensuring that the lateral flow strips were held away from high temperatures, Fig. S6). At the end of the reaction, the amplicons, generated within the LAMP chamber were eluted onto the lateral flow strips by pressing a blister pack/finger pump containing running buffer. The amplicons, labelled with both FITC and biotin ligand binding sites, contacted the end of the lateral flow devices and then were carried along the paper strips by capillarity, where they interacted with conjugation pads. This prototype HCV test device offers a cheap and user-friendly detection method together with the high sensitivity and specificity of the LAMP reaction.

To characterise the performance of the lateral flow detection devices, we tested a further 40 patient samples (20 HCV-positive and 20 healthy controls) on this platform and compared the results to our in-house qPCR assay (see 'Methods' for details). Results show no false positives (20/20 negatives detected correctly, Supplementary Table S4). Three clinically positive samples did not provide any Ct value on qPCR, indicative of either or both of low viral loads or degradation of the RNA from the additional

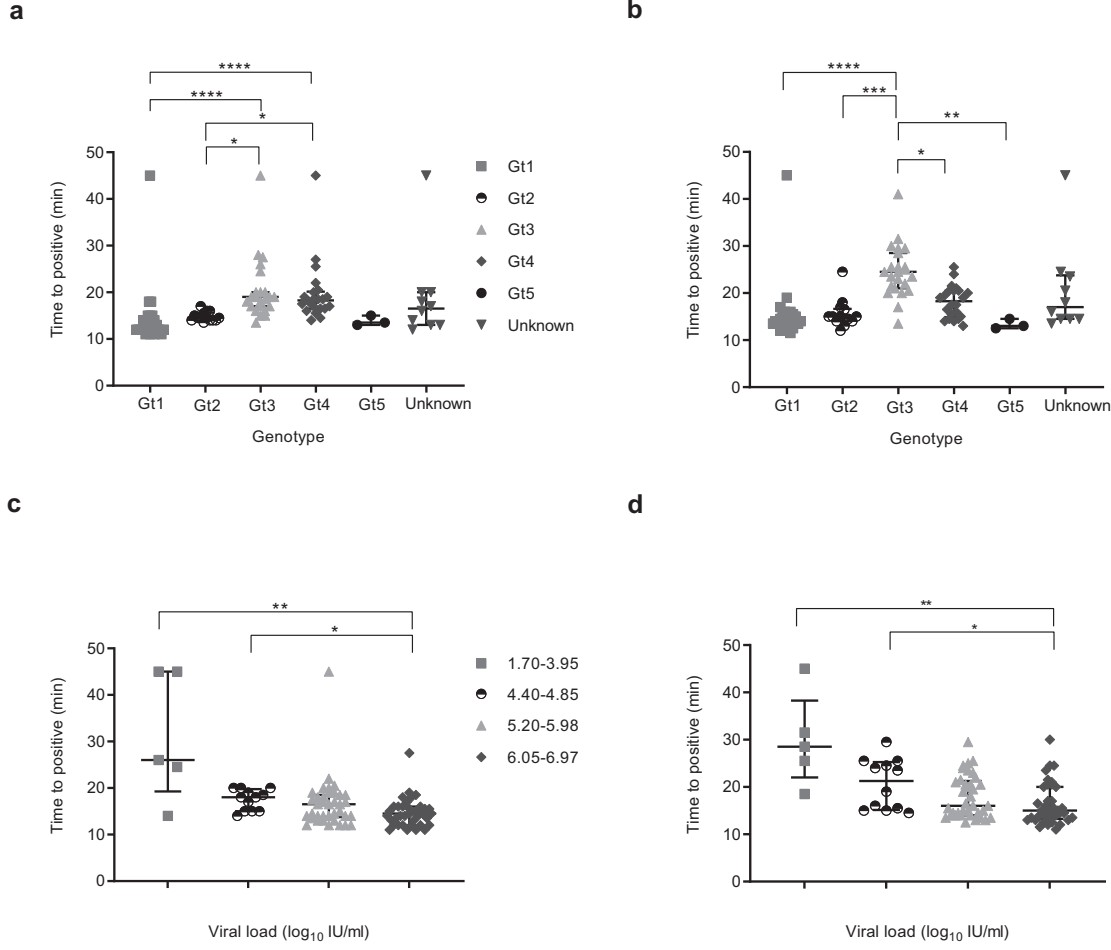

**Fig. 1 The effect of genotype and viral load on detection.** The central lines indicate median with interquartile range as error bars, and each point on the graph represents the mean of a sample run in duplicate. False negative samples were recorded as time to positive reaction at 45 min. Statistical analysis was performed using a one-sided non-parametric Kruskal-Wallis with Dunn's multiple comparisons test (one-sided). *$p \leq 0.05$, **$p \leq 0.01$, ***$p \leq 0.001$, ****$p \leq 0.0001$. **a** HCV RNA samples detection based on genotype (Gt): Gt1 (grey square, $n = 26$), Gt2 (white and black disc, $n = 14$), Gt3 (grey triangle, $n = 23$), Gt4 (grey lozenge, $n = 22$), Gt5 (black disc, $n = 3$), Unknown (grey inverse triangle, $n = 10$). Gt1 vs Gt3, $p < 0.0001$, Gt1 vs Gt4, $p < 0.0001$, Gt1 vs Unknown, $p = 0.1524$, Gt2 vs Gt3, $p = 0.0101$, Gt2 vs Gt4, $p = 0.0284$, Gt3 vs Gt5, $p = 0.1846$, Gt4 vs Gt5, $p = 0.2814$. All remaining groups had $p$ values of >0.9999. **b** HCV cDNA samples detection based on genotype; Gt1 (grey square, $n = 26$), Gt2 (white and black disc, $n = 14$), Gt3 (grey triangle, $n = 23$), Gt4 (grey lozenge, $n = 22$), Gt5 (black disc, $n = 3$), Unknown (grey inverse triangle, $n = 10$). Gt1 vs Gt3, $p < 0.0001$, Gt1 vs Gt4, $p = 0.0562$, Gt1 vs Unknown, $p = 0.3425$, Gt2 vs Gt3, $p = 0.0002$, Gt3 vs Gt4, $p = 0.0490$, Gt3 vs Gt5, $p = 0.0035$, Gt3 vs Unknown, $p = 0.3247$, Gt4 vs Gt5, $p = 0.3701$, Gt5 vs Unknown, $p = 0.5210$. All the remaining groups had $p$ values of >0.9999. **c** HCV RNA samples detection based on viral load: 1.7–3.95 (grey square, $n = 5$), 4.4–4.85 (black and white disc, $n = 11$), 5.2–5.98 (grey triangle, $n = 34$), 6.05–6.97 (grey lozenge, $n = 44$). 1.7–3.95 vs 5.2–5.98, $p = 0.1937$, 1.7–3.95 vs 6.05–6.97, $p = 0.0095$, 4.4–4.85 vs 6.05–6.97, $p = 0.0301$, 5.2–5.98 vs 6.05–6.97, $p = 0.4768$, 5.2–5.98 vs Unknown, $p = 0.4506$, 6.05–6.97 vs Unknown, $p = 0.0118$. All remaining groups had $p$ values of >0.9999. **d** HCV cDNA samples detection based on viral load; 1.7–3.95 (grey square, $n = 5$), 4.4–4.85 (black and white disc, $n = 11$), 5.2–5.98 (grey triangle, $n = 34$), 6.05–6.97 (grey lozenge, $n = 44$). 1.7–3.95 vs 5.2–5.98, $p = 0.0536$, 1.7–3.95 vs 6.05–6.97, $p = 0.0032$, 1.7–3.95 vs Unknown, $p = 0.5693$, 4.4–4.85 vs 6.05–6.97, $p = 0.0492$, 5.2–5.98 vs 6.05–6.97, $p = 0.9431$, 6.05–6.97 vs Unknown, $p = 0.7858$. All remaining groups had $p$ values of >0.9999. Source data are provided as a Source data file.

freeze-thaw cycles and longer storage. 14/17 positive samples were also identified correctly by the LAMP on lateral flow devices. Two of the false negative samples had Ct values above 30 for our in house qPCR, which is beyond our threshold for detection, indicating low viral loads. Consequently, only one sample (Ct 29) was negative for the lateral flow device, whilst being positive for qPCR, showing excellent agreement, in line with the results obtained for fluorescence read-outs and demonstrating the potential for this low-cost and user-friendly method.

## Discussion

In this study, we aimed to develop a cheap, sensitive and specific bedside test for HCV. Using optimised primers, we validated its potential as a future POC/bedside test in a double-blind study of clinical samples with varied viral loads and comprising all major genotypes, including a recently identified genotype 7 sample[22]. Both RT-LAMP and LAMP performed well in this study, exceeding the standards set out by WHO for POC tests[24]. The samples were diagnosed by comparison with the gold standard Abbott RealTime HCV PCR assay within a validated clinical reference laboratory and further compared with an in-house RT-PCR assay. A small number of samples that were not detected by RT-PCR were also missed by the LAMP assay, possibly due to low viral load.

The target product profile for diagnosis of HCV as recommended by WHO requires a minimum diagnostic specificity of >98% and sensitivity of >90–95% which was fulfilled by

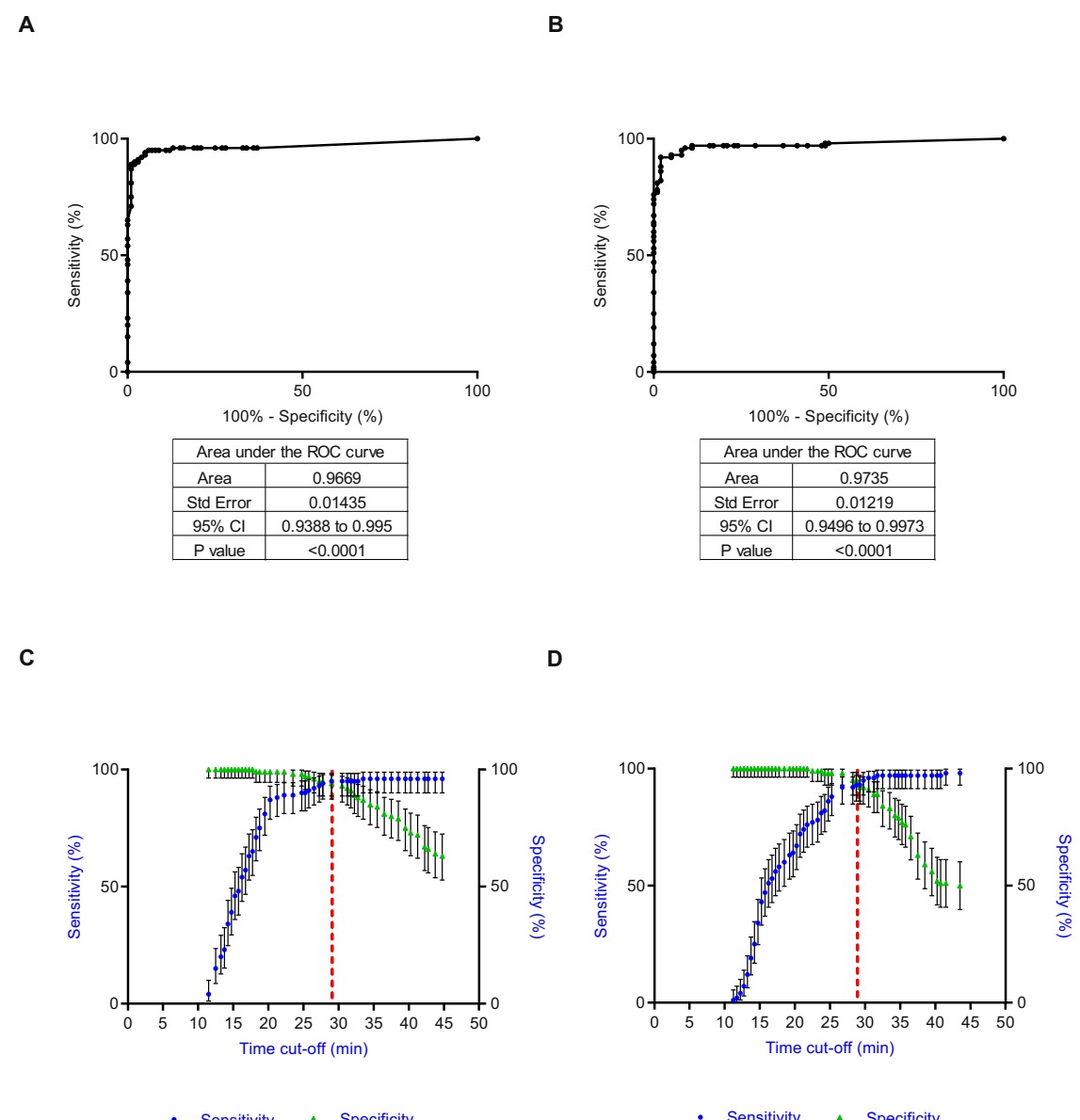

**Fig. 2 ROC curves for LAMP assays.** ROC curves were based on mean time to detection of 100 HCV positive and 100 HCV negative samples tested in a double-blind fashion. **A, B** ROC curve analysis, performed using the Clopper-Pearson Method (one-sided, no adjustments for multiple comparisons) for RT-LAMP and RNA samples (a) and for LAMP and cDNA samples (**B**). **C, D** Graphs showing sensitivity (circle) and specificity (triangle) at different time points in minutes for RT-LAMP (**C**) and LAMP (**D**). Error bars are 95% confidence intervals. The dashed line indicates the cut-off time in minutes, where sensitivity and specificity is optimal. Source data are provided as a Source data file.

RT-LAMP at a cut-off of <25 min[22]. However, the optimal analytical sensitivity of RT-LAMP assay, exceeding the WHO requirements, was reached at a cut-off of <30 min. The small number of false positive samples detected by LAMP versus RT-PCR could represent a slightly increased risk of cross-contamination or the formation of primer-dimer structures detected by LAMP. Overall, false positives did not occur commonly in this study, as in other HCV LAMP studies[17–19,25].

The analytical sensitivity of the assay at 40 min was ~2.6 $\log_{10}$ copies/reaction. In the original study from which we adapted our primers, the authors report a limit of detection of 50 IU/mL[17]. Both studies show that the assay falls well within the WHO's criteria for a point-of-care test (3000 IU/mL or below)[2].

The current diagnostic algorithm for HCV relies on testing for anti-HCV antibodies and later for HCV RNA to confirm active infection[1]. Although, rapid diagnostic tests for anti-HCV

antibody detection have been developed and FDA approved, they are still limited by the window period in early HCV infection when antibodies are undetectable[26]. Detection of current infection has to be confirmed by highly sensitive and specific HCV RNA detection assays including RT-PCR and those are still primarily limited by the high costs involved and incur a significant delay for patient management, which can be associated with poor treatment initiation outcomes[27]. The recent development of platforms such as GeneDrive and Cepheid Xpert, which are both CE-IVD certified, have improved the HCV diagnostic landscape with very high sensitivity and specificity, but require technical expertise and significant financial investment[7,8]. Both assays need a trained laboratory technician with at least one day of training to perform the assay. Furthermore, the cost of the GeneDrive platform is $5000 and $30–40 for each individual HCV test. In

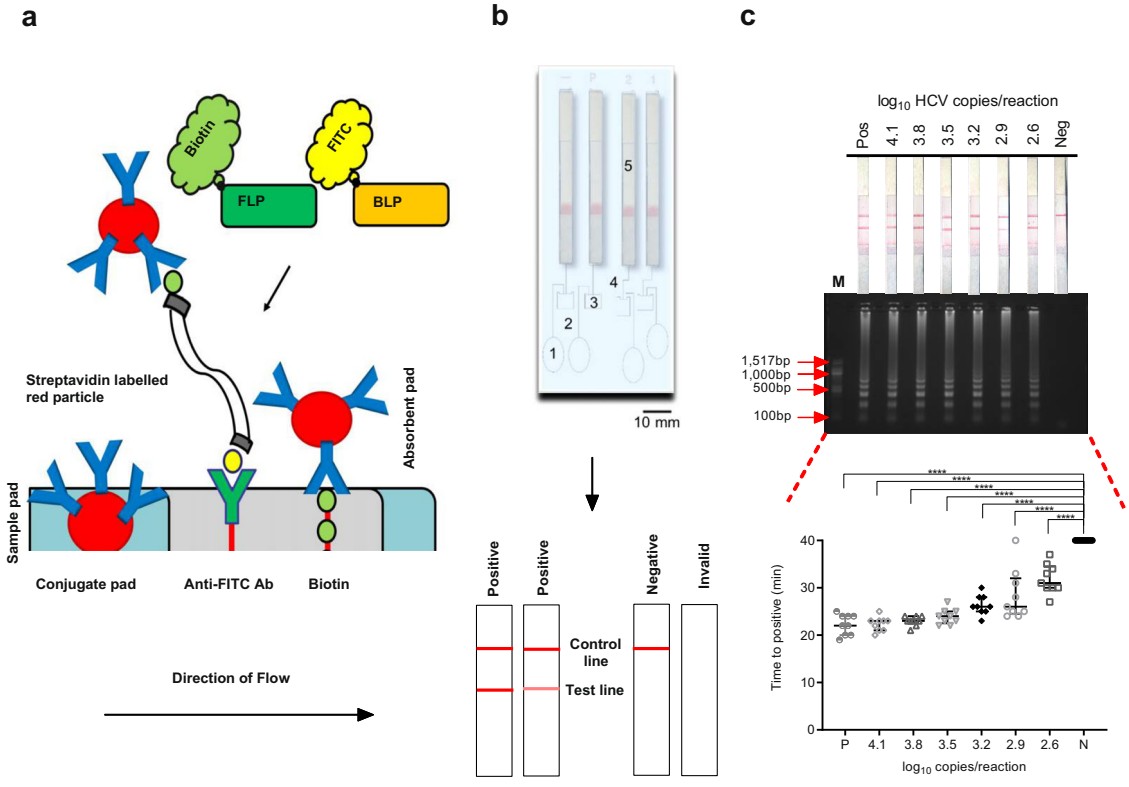

**Fig. 3 Lateral flow detection of HCV LAMP assay. a** The mechanism of the lateral flow strip[15]. Two primers, FLP and BLP, are pre-labelled with Biotin and FITC, respectively. The amplicon resulting from the LAMP reaction, contains both labelled primers as double-stranded DNA represented schematically by two gray lines. It is added onto the sample pad and moves towards the conjugate via capillary action. The streptavidin-labelled red particles bind with the Biotin (from the FLP primer) on the amplicon and together move towards the test line. The anti-FITC antibody (Ab) on the strip captures the amplicon via its FITC label (from the BLP primer) at the test line forming a band. Any unbound red particles move towards the control line where they are captured via the Biotin forming a second band. **b** The assembly and interpretation of the lateral flow devices. The device consists of the water chamber (1), connecting channels (2), four LAMP reaction chambers (3), channels (4) and lateral flow strips (5). Following the incubation period, two bands indicate a positive reaction, one band indicates a negative and no bands indicate invalid results. P—positive, N—negative, 1 and 2—sample in duplicate. **c** Analytical sensitivity of the lateral flow method (top panel) compared to gel electrophoresis (middle panel) and fluorescence over time (bottom panel). Serial dilutions of plasmid JFH1 replicon were made based on copy number per reaction ($\log_{10}$). Black lines indicate median with interquartile range ($n = 3$ biologically independent experiments, each with three technical replicates). The different symbols are for each dilution, to ease visualisation (grey and white disc—4.1, white lozenges—3.8, inverse triangles—3.5, black diamond—3.2, grey circles—2.9 and white squares—2.6 $\log_{10}$ copies/reaction, black disc is negative control—DI water). Statistical analysis for the fluorescence over time was performed using a parametric, one-way ANOVA. The F ratio = 55.56 and the degrees of freedom = 65. ****$p \leq 0.0001$, Pos—positive HCV control, Neg—no template control, M—100 bp NEB DNA ladder, C—control line, T—test line. Source data are provided as a Source data file.

comparison, the authors of this study have previously estimated the cost of a similar LAMP assay to be < $10[15].

The prototype lateral flow device developed in this study does not affect the sensitivity of the assay when compared to gel electrophoresis and fluorescence-based detection. Nucleic acid detection was carried out with an easy-to-interpret, pregnancy test-like visualisation output, requiring two labelled primers (Fluorescein isothiocyanate, FITC, labelled FLP and biotin, Bio, labelled BLP) specific to double-stranded HCV LAMP amplicons. Compared to our previous study[15], which focussed on the detection of plasmodium DNA, we demonstrate this capability with the amplification of RNA for the detection of HCV. This contrasts with previous HCV LAMP studies that have used SYBR green for colorimetric detection with non-specific detection of double-stranded DNA, which may result in higher numbers of false positive results[19].

In future work, we propose to integrate an RNA extraction system within the current lateral flow device. Candidate systems have already been described and have used LAMP directly from serum, urine and whole blood, for example in Zika virus detection, without significantly impairing assay performance[28–30]. Other studies have reported the use of paper microfluidics as a suitable RNA extraction method in conjugation with LAMP reactions including a recently published 30-second nucleic acid extraction protocol[31,32]. The incorporation of such methods with our highly sensitive and specific HCV LAMP assay would allow for a sample-to-answer test result within 60 min. A single visit would then be sufficient for diagnosis and initiation of treatment which could significantly improve the uptake of treatment in high risk groups including PWID and HIV infected men-who-have-sex-with-men by reducing loss to follow-up[33,34]. PWID have a high global prevalence of HCV infection with an estimated 10

million positive for anti-HCV antibodies in 148 countries[34,35]. Several studies have demonstrated that POC testing is cost-effective in high-risk groups as well as in countries with high HCV prevalence such as Egypt[36,37]. Since LAMP requires a stable temperature of 60–65 °C with no cycling, reactions can be established with minimal equipment requirements, for example by using a commercially available thermos flask, water bath or coffee mug[38]. Using such methods, LAMP performs well in the field for the diagnosis of malaria[15] and foot-and-mouth disease[14] amongst others.

HCV LAMP is suitable for diagnosis across a range of different viral loads and genotypes. Differences were noted in the time to detection of genotype 3 with the LAMP assay and genotypes 3 and 4 for RT-LAMP, although these were significantly improved by the alteration of the BLP primer in optimisation experiments and were detected in <30 min. Assay performance could be optimised further by altering primer sequences based on the genotypes most prevalent in the area in which diagnosis is taking place.

In conclusion, we have developed a highly sensitive and specific HCV LAMP assay, which exceeds WHO requirements for a diagnostic test. Our prototype lateral flow device is a promising tool for user-friendly POC testing, eliminating the need for follow up visits between diagnosis and initiation of treatment. Future studies should evaluate prospective testing in the field in patients receiving HCV treatment.

## Methods

**Clinical samples and standards**. 100 fully anonymised plasma samples from patients with chronic HCV infection were compared with 100 HCV-negative control samples. Samples, including healthy controls, were selected randomly from the West of Scotland Specialist Virology Centre (WoSSVC) via the Greater Glasgow and Clyde Health Bio-repository and the NHS Research Ethics Committee (REC), anonymised and processed from plasma at the MRC, Centre for Virus Research or from venous whole blood at the WoSSVC, Glasgow Royal Infirmary. Ethical approval was granted by the Greater Glasgow and Clyde Health Bio-repository and the NHS Research Ethics Committee, application number 606. All samples were obtained with informed consent, with no compensation. Samples were assigned a numerical sample ID (1–200 or 1–40) randomly (using the random number generator in Microsoft Excel 365) and processed from venous whole blood at the WoSSVC. The correspondence (positive, negative and viral load when relevant) was held by the WoSSVC. The samples were provided blinded for processing.

The HCV viral load was quantified using the Abbott RealTime HCV assay. The viral load groupings were defined by the WoSSVC according to common clinical practice. Sensitivity and specificity testing was carried out using a double-blind study design. Analytical sensitivity was determined by serial two-fold dilutions of JFH1 subgenomic replicon containing the *Guassia* luciferase gene (pSGR-HCV-JFH1-GLUC)[39]. The number of copies of replicon per reaction was calculated based on DNA mass and plasmid length on the NEB mass online calculator[40].

40 samples stored at −80 °C, from patients with HCV (n = 20) and negative controls (n = 20) were anonymised by WoSSVC staff. They were transported to the University of Glasgow on ice and stored at −80 °C until use. They were processed according to the procedure detailed below for the LAMP lateral flow devices and RT-PCR, in a double-blind fashion. The results were read independently by two assessors before unblinding.

**RNA extraction and cDNA synthesis**. RNA was extracted from 200 µL of plasma using the Agencourt RNAdvance Blood Kit (Beckman Coulter) on the automated KingFisher™ Flex Purification System, with a method adapted from the manufacturer's protocol. Samples were lysed with 300 µL of Lysis buffer and 30 µL of Proteinase K and the DNase stage was shortened to 5 min. Purified RNA was eluted in 25 µL of nuclease-free water. SuperScript III Reverse Transcriptase (Thermo Fisher Scientific) was used for complementary DNA synthesis from 11 µL of eluted RNA with random hexamers as per the manufacturer's instructions.

**HCV LAMP assay and product detection**. LAMP reactions were run in a final volume of 25 µL, containing 15 µL of ISO-001-RT Master Mix (Optigene, UK), 5 µL of target DNA/RNA, 0.8 µM of FIP and BIP, 0.4 µM of FLP and BLP, 0.2 µM of F3 and B3 primers and 0.4 µM of AP. Primer sequences are shown in Table 1.

Reaction mixtures were incubated for 40–45 min at 65 °C. Real time monitoring of LAMP reactions was performed on a 7500 Fast Real-Time PCR (RT-PCR) machine (Applied Biosystems) using the SYBR green setting. For end-point

detection, LAMP products were subjected to gel electrophoresis on a 1% agarose TAE gel dyed with ethidium bromide and the results detected under UV light. Additionally, LAMP products were visualised on nucleic acid detection strips (Ustar®, China). FLP and BLP were labelled with Biotin and FITC, respectively (Table 1), and the entire product was added to the strips and topped up with ~100 µL of nuclease free deionised water to enable the flow of the sample through the lateral strip using capillarity. The results were recorded within 10 min by the ESPON EXPRESSION 1680 Pro scanner with 300 dpi resolution. Images were processed in Microsoft PowerPoint 2016 software.

**Lateral flow devices**. Lateral flow devices containing chambers for LAMP reactions, finger pumps and nucleic acid detection strips were manufactured in-house on a laser cutter (Laserscript) from 2 mm-thick poly(methylmethacrylate) (Fig. 3b and Fig. S5a). Two single sided adhesive acetate films (MicroAmp Optical Adhesive Film; Thermo Scientific) were used to seal the devices and prevent evaporation during LAMP reactions. These devices are low-cost, estimated as follows: cartridge < $10c, detection strip $1, reagents (3 reactions (positive, negative, test)), $3. Total < $5.

**HCV RT-PCR assay**. An in-house RT-PCR with primer and probe sequences based on the 5′UTR (JFH1- primer 16; 5′-TCTGCGGAACCGGTGAGTAC-3′, JFH1-primer 17; 5′-GCACTCGCAAGCACCCTAT-3′, FAM probe; 6-FAM-AAAGGCCTTGTGGTACTG-MGB) was used for comparison, whilst the Abbott real-time RT-PCR assay was used as a gold-standard HCV detection reference assay. Each master mix consisted of 2× TaqMan Fast Universal Mix (Thermo Fisher Scientific), 18 µM forward and reverse primers, 5 µM probe and 1 µL cDNA template in a final volume of 10 µL. The run consisted of 20 s hold at 95 °C followed by 40 cycles of denaturation for 20 s at 95 °C and amplification for 30 s at 60 °C.

**Statistical analysis**. Normal distribution of data was determined in GraphPad Prism version 7 using D'Agostino & Person normality test. A parametric one-way ANOVA was used for normally distributed data and a non-parametric Kruskal-Wallis test with Dunn's multiple comparisons test was used for other data types. ROC curves were plotted on the same software. Data were considered significant if *p* value was less than or equal to 0.05.

**Reporting summary**. Further information on research design is available in the Nature Research Reporting Summary linked to this article.

## Data availability

The raw data, sequence alignment, gel blots and images that support the findings of this study are also available in University of Glasgow's Enlighten: Research Data with the identifier: https://doi.org/10.5525/gla.researchdata.1127. Source data are provided with this paper.

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

## Acknowledgements

The authors would like to Dr. Shantimoy Kar and Dr. Xiaoxiang Yan from the James Watt School of Engineering at the University of Glasgow for their inputs on the design of the lateral flow device. This study was funded in part by Engineering and Physical Sciences Research Council, Grant Number: EP/M508056/1, Medical Research Council Grant Numbers: G0801566, G0901213–92157, MC_PC_16045, MC_UU_12014/1, MR/K013491/1 and the Wellcome Trust, Grant Number: 102789/Z/13/A. We also acknowledge the support of NHS Research Scotland (NRS) Greater Glasgow and Clyde Biorepository.

## Author contributions

E.C.T., J.M.C. and J.R. supervised the project. A.B.S. determined the viral loads and genotypes of the HCV samples by the Abbott RealTime PCR. A.B.S. and R.G. provided and anonymised the samples for the double-blind studies. C.D. and S.R.S. extracted RNA and conducted cDNA synthesis of all samples for the double-blind study. W.M.W. performed the LAMP and in-house RT-PCR assays for the double-blind studies. W.W.M., A.G., G.X. and Z.Y. designed the lateral flow devices. W.W.M. and A.G. manufactured and assembled the lateral flow devices. W.W.M., P.J., J.M.C. and J.R. analysed the data. W.W.M. wrote the original manuscript. E.C.T., J.M.C. and J.R. provided revisions and all authors reviewed and edited the manuscript.

## Competing interests

The authors declare no competing interests.
