## [Peer Review File · Nature Communications]

Loop mediated isothermal amplification as a powerful tool for early diagnosis of hepatitis C virusReviewers' Comments:

Reviewer #1:

Remarks to the Author:

In the submitted manuscript "Loop mediated isothermal amplification as a powerful tool for early diagnosis of hepatitis C virus" (NCOMMS-21-12090), the authors aimed to develop a cheap, sensitive and specific point-of-care test for HCV detection. The authors highlighted that they developed a prototype device, comprising a LAMP amplification chamber and lateral flow detection strip, giving a result in <40 minutes with high sensitivity and specificity (>90%). This test is suitable for diagnosis across a range of different viral loads and genotypes (genotypes 1,2,3,4,5,6,7). This will create new opportunities for enhancing access to HCV test among individuals with HCV reside in low- and middle-income countries (LMICs).

However, there are some points that need to be clarified:

1. In the "Results" part (Optimization of HCV LAMP primers), the authors should add an explanation why did the authors decide to optimize only BLP primers and test with genotype 3 only. Did the new BLP primers test with other genotypes?
2. In the "Results" part (Clinical sensitivity and specificity), the authors mentioned that RT-LAMP and LAMP exhibited high sensitivity (96 and 97%) and specificity (91 and 90%). These results may come from fluorescence-based detection only, was this well correlated with lateral flow device? The authors should also provide the results of lateral flow device (sensitivity and specificity), comparing to standard laboratory reference techniques.
3. In the "Results" part (analytical sensitivity), the sensitivity should be evaluated with RT-PCR assay.
4. In the "Results" part (analytical sensitivity), the authors mentioned that the LOD was 2.6 log₁₀ copies/reaction (Figure 3). The LOD will be less than 2.6 log₁₀ copies/reaction if the plasmids containing HCV is more diluted.
5. Did the authors test the cross-reactivity with other blood-borne viruses?
6. Line 194, the authors should also add more recent reference (Hongjaisee S, Int J Infect Dis 2021).
7. In the "Methods" part, the authors should provide an explanation about clinical samples, especially HCV-negative control samples. What was their origin? Were there co-infection?

Best regard,

Reviewer

Reviewer #2:

Remarks to the Author:

The authors of this work present here a fieldable device for detection of hepatitis virus using reverse transcriptase loop mediated isothermal amplification (RT-LAMP) on simple microfluidic device.

The general structure of the paper reads well. The introduction is beautifully written, thorough–yet concise–and includes recent references. The experimental section is detailed. The results section is of good scientific quality and has no misconceptions or ambiguities.

I commend the authors for conducting a validation and verification study. This comparison to a gold

standard using clinical samples is typically done in industry. A large value creation exercise to all readers of this work. All results support the conclusions. This reviewer finds the improvement of 21% in positivity as significant evidence of the importance of this work. Time to detection was estimated using 10-sigma above the mean baseline fluorescence of the positive control. A cut-off value at less than ~25 minutes would allow achieving specificity as recommended by WHO at 98% but would decrease RT-LAMP sensitivity by about 5%. For analytical sensitivity and end-point detection, this work compared well to WHO requirements.

This work is a significant and timely addition to the field of paper-based diagnostics. Our current global needs in diagnostics, so brutally exposed by the COVID-19 pandemic, make this work even more pertinent and important to share with the readership of Nature Comms.

I unequivocally recommend publication of this manuscript with minor revisions, provided all the following comments are addressed:

1. Inconsistent spelling of word "pan-genotypic". Please correct.
2. Please elaborate more on why this characteristic pattern forms in gels. This refers to "characteristic smear with ladder-like banding pattern for all samples except the negative control".
3. Please measure what is the temperature on the strip itself using a thermocouple while the device is incubated for 30 minutes. A simple thermal profile would suffice.
4. I am not sure I understand the term "bathed". Please use different word.
5. Please provide with a rudimentary bill of materials for the total price of your in-house prototype. A rough estimate would suffice. For example: 1) Valve, \$XX, 2) Detection strips, \$XX, 3) Reagents, \$XX, Total \$XX
6. I read this paper from the authors of this submission with great interest (<https://doi.org/10.1073/pnas.1812296116>) and ref 14 in this manuscript. Please include a short paragraph highlighting the main improvements (if any) to the device since then.
7. What method did you use to anonymize patient samples? Please include a short sentence.

I am including my comments in line with the Word document, as tracked changes.

Detailed point-by-point response (in red) to reviewers' comments (in blue).

Please note that marked up versions of both the manuscript and supplementary information (changes marked in red in the text) are provided for ease of reference as additional files.

Reviewer 1

1. In the “Results” part (Optimization of HCV LAMP primers), the authors should add an explanation why did the authors decide to optimize only BLP primers and test with genotype 3 only. Did the new BLP primers test with other genotypes?

We thank the reviewer for this opportunity to detail our analysis further. We optimised only BLP, based on analysing an alignment file of over 200 sequences of major HCV genotypes and subtypes available in (Smith *et al.*, 2014 – reference 21). The majority of the observed mismatches were in the middle of the primers. We only noted one mismatch within the last base pair of the 3' end of the BLP, within genotype 3 sequences. Thus, we focused on the optimisation of BLP within genotype 3.

The new BLP was also compared to the original BLP performance on genotype 1 targets, which, as expected, did not result in significant differences in the assay performance - as there were no mismatches in either original or new BLP sequence within this genotype. We have added this information in the manuscript, as follows:

“Previously published LAMP primers were selected based on evidence of a low limit of detection from HCV RNA (50 IU/mL)¹⁷. In this study, the use of an additional accelerating primer (AP, Table 1) ensured improved sensitivity and specificity across several HCV genotypes. We analysed an alignment file of over 200 sequences of major HCV genotypes and subtypes²¹. The majority of the observed mismatches were in the middle of the primers (Fig. S1). We only noted one mismatch within the last base pair of the 3' end of the backward loop primer (BLP), within genotype 3 sequences. Focussing on genotype 3 and in order to further improve the previously published assay, we removed the cytosine mismatch at the 3' end of BLP. Additionally, a cytosine was added at the beginning of the primer, in order to conserve the primer melting temperature (Fig. S1 and Table 1). The removal of the cytosine mismatch and the subsequent use of the new primer improved the time to positivity by 21% in genotype 3 (Fig. S2). As a confirmation, the new BLP was also compared to the original BLP performance on genotype 1 targets, which, did not result in significant differences in the assay performance as there were no mismatches within this genotype.”

2. In the “Results” part (Clinical sensitivity and specificity), the authors mentioned that RT-LAMP and LAMP exhibited high sensitivity (96 and 97%) and specificity (91 and 90%). These results may come from fluorescence-based detection only, was this well correlated with lateral flow device? The authors should also provide the results of lateral flow device (sensitivity and specificity), comparing to standard laboratory reference techniques.

We agree with the reviewer that a full view of the performance of the test on the lateral flow devices will allow the reader further confidence in the approach. Unfortunately, the samples used previously for fluorescence evaluation are not available anymore (they are either depleted or have had too many freeze-thaw cycles that could impact on measurements).

We therefore acquired a new set of samples (ethical approval granted by the Greater Glasgow and Clyde Health Bio-repository and the NHS Research Ethics Committee, application number 606) and performed the analysis on the lateral flow devices.

Due to the impact of the COVID-19 pandemic, we were only able to acquire 40 samples, including 20 healthy controls and 20 from clinically-confirmed HCV patients, all previously frozen. We tested the samples in a double-blind fashion, repeated as quadruplexes (n=4) using both our in-house qPCR assay and the RT-LAMP assay with the lateral flow device. In all cases, the results were independently read by two operators before unblinding.

Results showed that 20 negative samples were characterised correctly by both methods. We also noted that 3 clinically-confirmed positive samples did not result in any Ct value on the qPCR, indicating potential degradation of the RNA (following freezing and storage). These were also negative on the lateral flow device. 14 out of the remaining 17 positive samples were also positive on the lateral flow device. Two of the false negative samples had Ct values above 30 for our in-house qPCR, which is beyond our threshold for detection, meaning that they were also false negatives for this assay, indicating low viral loads.

One sample (Ct 29) was negative for the lateral flow device, whilst being positive for qPCR, showing a very good agreement, in line with the results obtained previously for fluorescence read-outs. The results are provided in the text as follows and the data is available in Supplementary Table S4, reprised here for convenience. Experimental details have also been added to the Methods section to reflect these new experiments.

“To characterise the performance of the lateral flow detection devices, we tested a further 40 patient samples (20 HCV-positive and 20 healthy controls) on this platform and compared the results to our in-house qPCR assay (see Methods for details). Results show no false positives (20/20 negatives detected correctly, Supplementary Table S4). Three clinically positive samples did not provide any Ct value on qPCR, indicative of either or both of low viral loads or degradation of the RNA from the additional freeze-thaw cycles and longer storage. 14/17 positive samples were also identified correctly by the LAMP on lateral flow devices. Two of the false negative samples had Ct values above 30 for our in house qPCR, which is beyond our threshold for detection, indicating low viral loads. Consequently, only one sample (Ct 29) was negative for the lateral flow device, whilst being positive for qPCR, showing excellent agreement, in line with the results obtained for fluorescence read-outs and demonstrating the potential for this low-cost and user-friendly method.”

Methods Section:

“40 samples stored at -80°C, from patients with HCV (n=20) and negative controls (n=20) were anonymised by WoSSVC staff. They were transported to the University of Glasgow on ice and stored at -80°C until use. They were processed according to the procedure detailed below for the LAMP lateral flow devices and RT-PCR, in a double-blind fashion. The results were read independently by two assessors before unblinding.”

*Table S4. comparison between qPCR results and RT-LAMP on lateral flow devices for 40 clinical samples.*

| *RT-LAMP lateral flow* | ***qPCR*** | |
|---|---|---|
| | *Ct (mean)* | *Ct (Std)* |
| *positive* | *23.24* | *0.32* |
| *positive* | *24.08* | *0.07* |
| *positive* | *24.41* | *0.17* |
| *positive* | *24.83* | *0.11* |
| *positive* | *25.25* | *0.71* |
| *positive* | *25.83* | *0.38* |
| *positive* | *25.84* | *0.60* |
| *positive* | *26.17* | *0.01* |
| *positive* | *26.26* | *0.80* |
| *positive* | *26.39* | *0.36* |
| *positive* | *26.44* | *0.21* |
| *positive* | *26.56* | *0.04* |
| *positive* | *27.17* | *0.38* |
| *positive* | *28.71* | *0.23* |
| *negative* | *29.15* | *0.26* |
| *negative* | *32.13* | *0.23* |
| *negative* | *33.32* | *0.78* |
| *negative* | - | |
| *negative* | - | |
| *negative* | - | |
| *negative* | - | |
| *negative* | - | |
| *negative* | - | |
| *negative* | - | |
| *negative* | - | |
| *negative* | - | |
| *negative* | - | |
| *negative* | - | |
| *negative* | - | |
| *negative* | - | |
| *negative* | - | |
| *negative* | - | |
| *negative* | - | |
| *negative* | - | |
| *negative* | - | |
| *negative* | - | |
| *negative* | - | |
| *negative* | - | |
| *negative* | - | |
| *negative* | - | |

3. In the “Results” part (analytical sensitivity), the sensitivity should be evaluated with RT-PCR assay.

We evaluated the analytical sensitivity of the LAMP and RT-LAMP assays, with respect to the gold standard PCR techniques. The in-house RT-PCR assay was characterised elsewhere as a quantitative assay (Witteveldt *et al* 2009, Journal of General Virology). We have added a statement to clarify this in the Methods section, as follows:

‘The performance of HCV LAMP using RNA samples (RT-LAMP) directly and HCV LAMP using cDNA as template were compared with a highly sensitive in-house HCV RT- PCR by genotype and viral load (Tables 2 and 3). The in-house RT-PCR assay has been characterised previously as a quantitative assay (qRT-PCR²³)’

4. In the “Results” part (analytical sensitivity), the authors mentioned that the LOD was 2.6 log₁₀ copies/reaction (Figure 3). The LOD will be less than 2.6 log₁₀ copies/reaction if the plasmids containing HCV is more diluted.

The WHO recommends that the limit of detection of a HCV point-of-care test should be ≤3000 IU/ml. As discussed in the manuscript, the 2.6 log₁₀ copies/reaction is equivalent to a detection threshold of ca. 398 copies/reaction, below the recommended limit of detection. We agree with the referee that the LOD will be below this value, and we have added this qualification as follows:

‘The lower limit of detection was the same for each method, below 2.6 log₁₀ copies/reaction (broadly equivalent to a detection threshold of 398 copies/reaction, which is lower than the recommended ≤3000 IU/mL limit of detection by WHO)².’

5. Did the authors test the cross-reactivity with other blood-borne viruses?

We do not have ethical approval to comment on co-infections. The samples were selected at random and anonymised by the West of Scotland Specialist Virology Centre.

6. Line 194, the authors should also add more recent reference (Hongjaisee S, Int J Infect Dis 2021).

We thank the reviewer for this useful reference and have added it to the text, as follows:

‘Recently, LAMP assays have also been developed for HCV detection in centralised facilities. Colorimetric outputs have also been developed to enable the detection without bulky optical instrumentation with excellent performance for some genotypes (e.g. Hongjaisee S. et al.¹⁶ showed 100% sensitivity with genotype 6). However, these assays showed limited diagnostic performance when used with varied genotypes or low viral loads¹⁷⁻¹⁹. Their application as POC tests is thus restricted, especially in LMICs²⁰, where a wide variation in the genetic diversity may lead to a less-good efficacy of tests.’

7. In the “Methods” part, the authors should provide an explanation about clinical samples, especially HCV-negative control samples. What was their origin? Were there co-infection?

We thank the reviewer for this opportunity to provide clarifications on the samples. As mentioned in comment 5., we did not have ethical approval to study co-infections. We have added the following clarifications in the Methods section:

‘Samples, including healthy controls, were selected randomly from the West of Scotland Specialist Virology Centre (WoSSVC) via the Greater Glasgow and Clyde Health Bio-repository and the NHS Research Ethics Committee (REC), anonymised and processed from plasma at the MRC, Centre for Virus Research or from venous whole blood at the WoSSVC, Glasgow Royal Infirmary.’

Reviewer 2

1. Inconsistent spelling of word “pan-genotypic”. Please correct.

We have corrected all instances to ‘pan-genotypic’

2. Please elaborate more on why this characteristic pattern forms in gels. This refers to “characteristic smear with ladder-like banding pattern for all samples except the negative control”.

The HCV LAMP reaction uses 7 primers. It is initiated by the F2/B2 part of inner primers (FIP/BIP). The F3/B3 primers then bind upstream from this region, causing the displacement of the initial strand. This leads to bands for the initial amplification steps, generating products with simple loops (at 274 bp representing the region between F3/B3). This loop structure has multiple initiation sites for amplification for the inner primers (FIP/BIP), the loop primers and the accelerating primer (AP). As the reaction proceeds, long concatemers are created, with random termination, resulting in accumulation of double-stranded DNA with different sizes. We have added these details in the text as follows:

‘The gel electrophoresis revealed a characteristic smear with ladder-like banding pattern for all samples except the negative control. The sizes of the initial bands (274 bp) correlate with the region between F3/B3 representing the initial stem loop formation of the reaction. As the reaction proceeded, larger constructs were created, with random termination, generating a other bands with a smear-like pattern^{11,17,12}. The results correlated with the nucleic acid detection strips.’

3. Please measure what is the temperature on the strip itself using a thermocouple while the device is incubated for 30 minutes. A simple thermal profile would suffice.

We have measured the temperature on the device using a thermocouple and a representative profile is provided in Figure S6, reprised here for convenience. The temperature once reaching 65°C was stable at 65.8°C (+/- 0.2°C, standard deviation)

Fig. S6. Temperature monitoring during the LAMP reaction in the heater device (Fig. S5b). The temperature was measured using thermocouples placed in different positions in device (a). (b) Results show a stable temperature at the LAMP reaction value of 65.8°C (+/- 0.2°C, standard deviation), whilst it decreases rapidly away from it to reach close to room temperature (which was 20°C) for measurements performed on the strips (31.3°C +/- 0.7°C). The higher fluctuations for the temperatures measured in the channels and the strip can be explained by the fact that these areas are exposed to the room airflow.

4. I am not sure I understand the term “bathed”. Please use different word.

We modified to ‘contacted’:

‘The amplicons, labelled with both FITC and biotin ligand binding sites, contacted the end of the lateral flow devices and then were carried along the paper strips by capillarity’

5. Please provide with a rudimentary bill of materials for the total price of your in-house prototype. A rough estimate would suffice. For example: 1) Valve, \$XX, 2) Detection strips, \$XX, 3) Reagents, \$XX, Total \$XX

We thank the reviewer for this piece of advice. We are, in general, reluctant to provide cost/prices, as these vary significantly not only within economics cycles (cf cost of primers during the pandemic), but also with the scale of production (from lab to the factory). However, we understand the need to qualify the ‘low-cost’ of the device and provide estimates for a lab-based manual production (for small quantities <100), in the Methods section as follows:

Cartridge <10c, detection strip \$1, reagents (3 reactions (positive, negative, test)), \$3. Total < 5\$.

6. I read this paper from the authors of this submission with great interest (<https://doi.org/10.1073/pnas.1812296116>) and ref 14 in this manuscript. Please include a short paragraph highlighting the main improvements (if any) to the device since then.

We thank the reviewer for their interest in our previous work. We highlight that the key difference in this current work lies with the implementation of integrated reverse transcription with the amplification. In our previous work, detecting malaria, we used LAMP to amplify DNA directly, whereas here HCV, as an RNA virus, requires the reverse transcription of RNA to DNA before amplification. This was enabled in a similar timeframe as for DNA only, under 45min. We have added a short statement to emphasize this technical feature, as follows:

“Compared to our previous study¹⁵, which focussed on the detection of plasmodium DNA, we demonstrate this capability with the amplification of RNA for the detection of HCV.”

7. What method did you use to anonymize patient samples? Please include a short sentence.

We added a short statement in the methods section (merged with changes in response to review 1) as follows:

‘Samples were assigned a numerical sample ID (1-200 or 1-40) randomly (using the random number generator in Microsoft Excel 365) and processed from venous whole blood at the WoSSVC. The correspondence (positive, negative and viral load when relevant) was held by the WoSSVC. The samples were provided blinded for processing.’

Reviewers' Comments:

Reviewer #1:

Remarks to the Author:

The authors provide a clearer explanation. They have addressed all the comments and revised accordingly.

Reviewer #2:

Remarks to the Author:

Dear Authors,

I appreciate your carefully addressing all my comments.

Your work is significantly improving the field, and ideal for the readership of Nature Comms.

I recommend this paper for publication in this journal.